# Feature Contrastive Learning for No-Reference Segmentation Quality Evaluation

Xiaofan Li [ID], Bo Peng *[ID] and Zhuyang Xie

School of Computing and Artificial Intelligence, Southwest Jiaotong University, Chengdu 610031, China; xfl@my.swjtu.edu.cn (X.L.); zyxie@my.swjtu.edu.cn (Z.X.)
* Correspondence: bpeng@swjtu.edu.cn; Tel.: +86-132-0818-2600

**Abstract:** No-reference segmentation quality evaluation aims to evaluate the quality of image segmentation without any reference image during the application process. It usually depends on certain quality criteria to describe a good segmentation with some prior knowledge. Therefore, there is a need for a precise description of the objects in the segmentation and an integration of the representation in the evaluation process. In this paper, from the perspective of understanding the semantic relationship between the original image and the segmentation results, we propose a feature contrastive learning method. This method can enhance the performance of no-reference segmentation quality evaluations and be applied in semantic segmentation scenarios. By learning the pixel-level similarity between the original image and the segmentation result, a contrastive learning step is performed in the feature space. In addition, a class activation map (CAM) is used to guide the evaluation, making the score more consistent with the human visual judgement. Experiments were conducted on the PASCAL VOC2012 dataset, with segmentation results obtained by state-of-the-art (SoA) segmentation methods. We adopted two meta-measure criteria to validate the efficiency of the proposed method. Compared with other no-reference evaluation methods, our method achieves a higher accuracy which is comparable to the supervised evaluation methods and partly even exceeds them.

**Keywords:** segmentation quality evaluation; contrastive learning; meta-measure





## 1. Introduction

In recent years, image segmentation has been widely applied in the fields of autonomous driving [1,2], remote sensing image processing [3,4], medical image processing [5,6], etc., which has had a large influence on its excellent performance in visual tasks. Segmentation quality evaluation refers to a quantitative evaluation of the segmentation quality, so that the evaluation result can be used to measure the performance of segmentation algorithms and guide the adjustment of algorithm parameters. Furthermore, evaluation criteria can be used as a standard for designing a good segmentation algorithm. In short, segmentation quality evaluation is an essential process for image segmentation. In contrast to quality evaluation [7,8], which evaluates the quality of the image itself (distortion, blur, etc.), segmentation quality evaluation is more concerned with assessing how well the segmentation image extracts the object of interest from the original image.

There are full-reference evaluation methods, such as Mean Intersection over Union (MIoU) [9,10], Mean Pixel Accuracy (MPA) [9,11], F-Measure [12], Probabilistic Rand Index (PRI) [13], and Dice coefficient (Dice) [14]. These methods produce scores by calculating the similarities or the differences between the segmentation and the ground truth (reference image). However, there are several problems with such kinds of methods. Firstly, they require the ground truth labels as a reference, which requires a lot of manual effort regarding pixel-level labeling and is unable to capture the various semantic meanings of real-world objects. Secondly, they only evaluate the spatial relationship between the segmentation and the ground truth (e.g., the region areas or the boundary locations), which does not utilize

image cues in the evaluation process, resulting in an inconsistent evaluation with human visual standards.

No-reference methods can ease the dependence on the reference in the practical application process, such as ground truth, and therefore have become a promising solution for online segmentation evaluation tasks. Traditional no-reference methods [15–17] mainly use low-level image features (e.g., textures and colours) for evaluation; however, they are inefficient in semantic segmentation scenarios. It has been widely observed that learning the semantic information of the objects requires a large number of samples from class-specific objects, which is difficult to obtain in traditional evaluation methods. Intuitively, a good evaluation method should extract the meaningful semantic information from the image and distinguish the quality of segmentations based on this. Exploring the relationship between the original image and the segmentation result and quantifying the distance between segmentations are two important methods for a reasonable no-reference evaluation.

Meta-measures [18] are designed to measure the appropriateness of evaluation methods, and contain a series of principles for a general purpose evaluation. They are usually based on the ability to distinguish between segmentation images, e.g., identifying which segmentation is produced from a different original image, or on the ability to distinguish those with a higher quality. This provides a natural way to evaluate the performance of different segmentation measures and is designed independently from these measures.

In light of the superior performance of deep convolutional neural networks (CNNs) in feature representation, we propose a feature contrastive learning method for no-reference segmentation quality evaluations. Contrastive learning is a popular topic in computer vision research with many applications such as person re-identification [19], image matching [20], and visual tracking [21]. Contrastive learning is a type of self-supervised learning [22] that learns by comparing the commonalities and differences between pairs, which can reveal more about the relationships between parts of the data than other learning methods. More interestingly, the concept of comparing pair candidates coincides with the principles of meta-measures. Therefore, we integrate contrastive learning into the segmentation evaluation task and propose a CNN framework for no-reference segmentation evaluations.

The proposed framework does not perform contrastive learning directly because the amount of data required for direct contrastive learning is too large. Instead, it first learns the pixel-level similarity between the original image and the segmentation image and extracts the feature space. Next, a Siamese network is constructed. This network has two branches that share parameters. Each branch is based on a two-channel network [23] and loaded with preliminary learning parameters. The segmentation images are grouped into pairs according to different qualities, and the concatenated original images are input into the Siamese network for feature extraction. After that, a contrastive learning module is designed to learn feature similarities by calculating the extent to which this pair is related to the original image. In the prediction phase, in order to simulate the human perception process, we add a class activation map (CAM) [24] to the network, making the score more weighted towards the regions of attention.

To verify the effectiveness of the proposed framework, we produce 17,774 segmentations from the Pascal VOC2012 dataset using four SoA algorithms, which include 8887 well-segmented images and 8887 poorly segmented images. Two meta-measure criteria, including the Swapped-Image SoA Discrimination (SISD) [18] and the newly proposed Corresponding Image SoA Discrimination (CISD), are used to compare our method with both the no-reference and reference evaluation methods. The comparison results demonstrate the effectiveness of our method.

The main contributions of this paper are two-fold:

1. We present a new no-reference segmentation evaluation framework. It involves deep semantic information not covered by previous no-reference methods, demonstrating the substantial performance benefits of our method. We propose a prediction network by

using contrastive learning and a CAM module in segmentation quality evaluation from the perspective of learning.

2. We construct a new segmentation evaluation dataset and design a new meta-measure: CISD. The CISD together with the SISD criterion is used to test various segmentation evaluation methods on the new dataset, providing a reference for segmentation validation and analysis. Extensive experiments are performed to validate the efficiency of our evaluation framework, including preliminary pixel-level learning results, intervals of score distribution, examples of actual evaluation scores, etc.

The rest of the paper is organized as follows. Section 2 introduces the related work on segmentation quality evaluations. Section 3 describes the problem and presents the proposed evaluation framework, which consists of three important modules: pixel-level similarity learning, feature contrastive learning, and score adjustment with a CAM. Section 4 demonstrates the experimental settings, the dataset construction, the meta-measure methods, and the experimental results. Section 5 concludes the paper and discusses the future work.

## 2. Related Work

In this section, we review some recent advancements in four related topics: segmentation quality evaluation, metric learning, contrastive learning, and class activation maps.

### 2.1. Segmentation Quality Evaluation

The evaluation methods can be divided into two categories: full-reference and no-reference methods. Full-reference methods require the segmentation ground truth as a reference. A no-reference method does not require any additional content in the evaluation process. It is important to note that the no-reference method is not unsupervised and it does not matter what work was performed before the method was applied, including supervised learning.

The commonly used full-reference methods include Mean Pixel Accuracy (MPA), Precision [25], Recall [25], F-measure, Mean Intersection over Union (MIoU), Probabilistic Rand Index (PRI), Variation of Information (VI) [26], etc. MPA computes the ratio of correct classifications at the pixel level. The PRI is evaluated in terms of the Rand index, but has the same performance as the MPA. Precision computes the correct predictions of foreground in the prediction results. Recall computes the correct predictions of foreground in the truth. F is the weighted average of Precision and Recall. MIoU computes the average of the ratio of intersection and union for all categories. Dice is a set similarity measure function and is the ratio of the two-fold intersection to the sum of the segmentation image and the ground truth. VI computes the ratio of non-intersections from the information difference perspective.

No-reference methods mainly include E [15], Q [16], F [17], F' [27], Zeb [28], Ecw [29], etc. The evaluation method E uses region entropy as the measure of intra-region uniformity, which measures the entropy of pixel intensities within each region. Ecw uses the intra-region visual error to evaluate the degree of under-segmentation and uses the inter-region region visual error to evaluate the degree of over-segmentation. Zeb is based on the internal and external contrasts of the regions measured in the neighborhood of each pixel. F, F', and Q are based on the average squared color error of each region. These methods still use traditional machine learning methods, which are far less accurate and stable than supervised no-reference methods.

Recently, no-reference evaluation methods based on learning have been proposed. QualityNet [30] treats the evaluation as a regression problem, learning a regression value directly based on deep learning. However, its performance does not exceed that of IoU [31] because it uses IoU as a benchmark to obtain labels, which is a sub-optimal method for IoU. It can be taken as a simplification of the MIoU using two-class segmentation scenarios. These traditional no-reference methods and deep-learning-based methods are not designed to evaluate semantic segmentation scenarios that involve explicit object classes.

## 2.2. Metric Learning and Contrastive Learning

Metric learning is also known as similarity learning. In order to measure the similarity between samples, metric learning techniques can extract commonalities or determine differences between samples [32]. In the fields of face recognition [33,34], object tracking [35,36], one-shot learning [37], few-shot learning [38,39], and contrastive learning [40,41], many research works are based on metric learning. Metric learning can also be divided into traditional metric learning and deep metric learning.

Traditional metric learning is mainly a metric used by traditional machine learning methods such as K-Nearest Neighbor (KNN), Support Vector Machine (SVM), etc. They directly use fixed metric functions for metrics, such as the Euclidean distance, the Marxian distance, etc. The performance of these methods is limited.

Currently, there are various works on deep metric learning, which involve (i) extracting features using deep learning methods and then using metrics such as the Euclidean distance, (ii) deep learning methods for extracting features and measuring similarity, and (iii) direct similarity measures using deep learning, which only use inter-sample similarity features and do not extract features for a single sample. For example, [36] uses (i) to implement object tracking with a Siamese network, [39] uses (ii) to implement few-shot learning with a relation network, and [23] uses (iii) to compute the image similarity with a two-channel network, which can be applied in areas such as image retrieval [42] and image fusion [43].

Contrastive learning [44] is a kind of self-supervised discriminative method and its core is metric learning. It aims at grouping similar samples closer together and diverse samples far from each other. Self-supervised learning is another learning strategy other than traditional learning methods and has become popular in recent years. It improves the feature extraction capability by designing proxy tasks for the representational properties as the supervised information [45]. Self-supervision is reliable because it still uses labels which come from the properties of the data itself.

Corresponding to the proxy task of self-supervised learning, the key to contrastive learning is the design of positive and negative sample pairs. In contrastive learning, a common network is the Siamese network, which is used in current advanced methods such as SimCLR [40] and MoCov2 [41].

## 2.3. Class Activation Map (CAM)

A CAM is a heat map generated based on the feature space during network learning that can reflect the clues of deep features such as targets classes and can also be applied in weakly supervised localization and segmentation fields. Common methods to obtain class activation maps include CAM [46], Grad-CAM [47], Grad-CAM++ [48], Smooth Grad-CAM++ [24], etc. For these networks, the CAM saliency regions are highly correlated with human attention regions.

The CAM method uses the global average pooling operation to obtain the weight of each layer's feature map. Then, the weighted average of the depth direction of the feature map is taken as the class activation map. Grad-CAM incorporates the ReLU [49] operation when obtaining the weight, and builds on this to obtain the weights of each feature layer in the depth direction with backward gradient propagation, assigning different weights to each feature layer rather than using the global average as the weight. However, this method is problematic when there are multiple targets of the same class in the sample data. To optimize this, Grad-CAM++ still uses the ReLU operation when obtaining the weights, but different weight gradients are assigned to each category of weights. On this basis, Smooth Grad-CAM++ introduced the smooth grad to make the obtained class activation map clearer.

## 3. Proposed Framework of Feature Contrastive Learning

The goal of no-reference segmentation quality evaluations is to evaluate the quality of segmentation results without the ground truth. Most existing no-reference methods suffer from performance deficiencies, where the evaluation results do not conform to human vision and are difficult to apply in the field of semantic segmentation.

To address these issues, we explore the use of a learning-based method to better utilize the relationship between the original image and the segmentation image. A novel method is proposed for no-reference segmentation evaluation with contrastive learning.

Contrastive learning complies with the concept of segmentation evaluation meta-measuring in comparative image pairs. However, contrastive learning has a lower learning content, it requires a large amount of data, and is difficult to fit. Therefore, our framework first performs pixel-level similarity learning [50], which allows the network to understand the pixel-level relationship between the original image and the segmentation image. Then, contrastive learning is performed in the feature space. By comparing the differences in segmentation images of different quality, the network learns the global relationship between the segmentation and original images at the image level.

In next section, we will describe the details of the proposed method, which includes pixel-level similarity learning and feature contrastive learning processes. Then, the prediction phase is introduced, focusing on how to incorporate a class activation map (CAM) to generate the final evaluation score.

### 3.1. Pixel-Level Similarity Learning

With the limitation of the number of data, direct use of contrastive learning easily encounters problems such as overfitting. Instead of using contrastive learning directly, pixel-level similarity learning is performed in the preliminary learning with a two-channel network. In contrast to some methods [51,52], which uses traditional algorithm metric similarity, such as the euclidean distance, our method directly uses deep learning to measure the similarity, which only extract inter-sample similarity features and does not extract features for a single sample.

As shown in Figure 1, firstly, the original image and the segmentation image are concatenated as a $H \times W \times (C * 2)$ matrix for the network input. $H$ and $W$ represent the height and the width and $C$ is the channel. Then, the similarity features are obtained after the Resnet50 [53] with an upsample module. They are sent into the pixel-level decision layer to obtain a $H \times W$ feature map. Following some classical deep learning methods [54,55] validated with the Pascal Voc2012 dataset, we set $H = W = 320$ and $C = 3$.

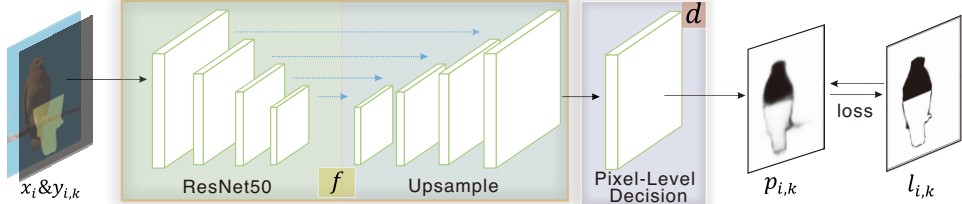

**Figure 1.** Pixel-level similarity learning. The original image ($320 \times 320 \times 3$) and the segmentation image ($320 \times 320 \times 3$) to be evaluated are combined as input ($320 \times 320 \times 6$), and a feature space of size $320 \times 320 \times 256$ is generated by the function $f$ (the downsampling module of ResNet50 and the upsampling module). Then, the pixel-level decision function d (a $1 \times 1$ convolution module) is used to generate pixel-level similarity matching results ($320 \times 320$).

Specially, we define a dataset $\mathcal{D} = \{x_i, \{y_{i,k}\}_{k=1}^{K}, z_i\}_{i=1}^{N}$, composed of $N$ independent and identically distributed training samples, where $x_i \in \mathcal{X}$ is the original image with dimensions of $320 \times 320 \times 3$. $\{y_{i,k}\}_{k=1}^{K} \in \mathcal{Y}$ refers to the set of segmentation images corresponding to $x_i$ and $z_i \in \mathcal{Z}$ is the ground truth of $x_i$. $y_{i,k}$ and $z_i$ have the same dimensions as $x_i$.

For original image $x_i$ and the segmentation image $y_{i,k}$ to be evaluated, label $\mathcal{L}$ is defined as in Equation (1).

$$l_{i,k} \in \mathcal{L}, l_{i,k} = y_{i,k} \odot z_i \tag{1}$$

where $\odot$ means the Inclusive-OR operation. $l_{i,k}$ is a similarity matching map with dimensions of $320 \times 320$.

With the determined input set as $(\mathcal{X}, \mathcal{Y})$ and the label set as $\mathcal{L}$, a function pair $\{f, d\}$ is used, where $f : (\mathcal{X}, \mathcal{Y}) \to \mathcal{F}$ is an upsample module to extract features and $d : (\mathcal{F}) \to \mathcal{P}$ is a decision function for pixel-level similarity. $\mathcal{F}$ represents the feature space with a size of $320 \times 320 \times 256$ and $p_{i,k} \in \mathcal{P}$ represents the prediction result with a size of $320 \times 320$. The number of channels is set empirically as 256.

In the proposed method, a fully convolutional network (FCN) [54] structure, $f$, is used as the upsample module to extract features. A simple convolutional module is used as the decision function, $d$. Corresponding to pixel-level similarity learning, we use the Mean Pixel Accuracy (MPA) as a loss function to learn the pixel-level relationship between the original image and the segmentation image, Equation (2).

$$Loss_{MPA} = 1 - \frac{\sum_{h=1}^{H} \sum_{w=1}^{W} MPA_{h,w}}{H * W}. \tag{2}$$

### 3.2. Feature Contrastive Learning with a Siamese Network

In most of the literature, there is no direct correlation between evaluation scores and the meta evaluation principle [14]. In this work, we attempt to integrate these two factors in the evaluation by using a contrastive learning strategy. The basic idea is to construct a contrastive loss function according to the meta-measure criteria for learning and then produce the evaluation scores. In particular, we only carry out contrastive learning for the feature maps extracted in the preliminary learning stage. Unlike other contrastive learning methods, which are mainly applied to upstream tasks, contrastive learning in this work is applied to downstream tasks to obtain the evaluation scores.

As shown in Figure 2, the similarity learning network is extended to a two-branch structure which makes a Siamese network. The original image concatenated with the positive segmentation is input to the upper branch and concatenated with the negative segmentation from the lower branch to extract features by $f$. After that, a contrastive module is constructed for contrastive learning. It contains a separate convolution operation $g$ for obtaining the upper and lower branch global features, and these features are averaged to obtain the scores. A contrastive loss is calculated between the upper and the lower branch scores. We adopt the contrastive principle: the similarity score of the upper branch should be higher than that of the lower branch.

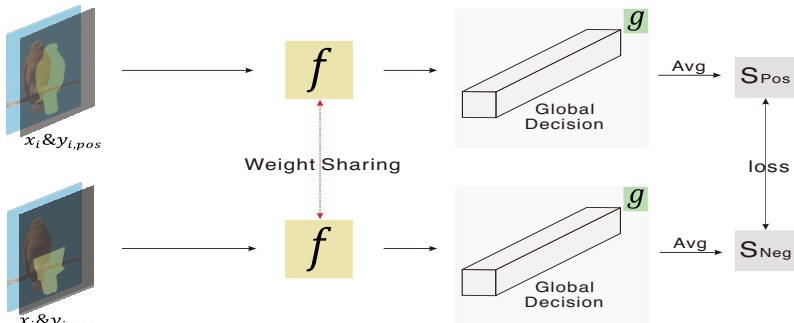

**Figure 2.** Feature contrastive learning with a Siamese network. The original image is combined with well-segmented results as the upper branch and combined with poorly segmented results as the lower branch. The functions $f$ are the same as for Figure 1. $f$ contains a ResNet50 and an upsampling module and is loaded with parameters for pixel-level similarity learning. For the output feature space ($320 \times 320 \times 256$), the global decision layer $g$ uses a large convolution kernel ($320 \times 320$) to produce a $1 \times 1 \times 256$ feature vector with its average value as the score. This learning process is image-level learning, which is different from pixel-level learning. Therefore, $g$ can be regarded as a global decision function.

Specifically, for the original image $x_i \in \mathcal{X}$, we select positive segmentation $y_{i,pos} \in \mathcal{Y}_{pos}$ ($pos \in [1, K]$) and negative segmentation $y_{i,neg} \in \mathcal{Y}_{neg}$ ($neg \in [1, K]$) from $\{y_{i,k}\}_{k=1}^{K}$.

The positive sample can be a good segmentation of the state-of-the-art algorithms or the segmentation ground truth, and the negative sample can be the poor segmentation of the algorithms or the segmentation from a different image.

The upper branch input pair is $\{\mathcal{X}, \mathcal{Y}_{pos}\}$ and the lower branch input pair is $\{\mathcal{X}, \mathcal{Y}_{neg}\}$. A function pair $\{f, g\}$ replaces $\{f, d\}$, where $f$ is the feature-extracting function for pixel-level similarity and $g$ is the global decision function. It is worth noting that $d$ is a pixel-level similarity decision function, it only learns pixel-level relationships and the output dimensions are $320 \times 320$, while $g$ is image-level learning, it is a global similarity decision function and the output is a vector with dimensions $1 \times 1 \times 256$. In this phase, the function $f$ is not used for back-propagation.

For the upper branch input pair $\{\mathcal{X}, \mathcal{Y}_{pos}\}$ and the lower branch input pair $\{\mathcal{X}, \mathcal{Y}_{neg}\}$, the features $\mathcal{F}_{pos}$ and $\mathcal{F}_{neg}$ are extracted separately using the function $f$. The function $g$ is used to obtain an average to obtain the upper and lower scores, $\mathcal{S}_{pos}$ and $\mathcal{S}_{neg}$. The function $g$ contains a convolution operation. Among them, $\mathcal{S}_{pos}$ and $\mathcal{S}_{neg}$ are the evaluation scores of good and poor segmentation outputs over the network, and they are a pair of scalar values. Based on the contrastive learning principle, a hyper parameter $\alpha$ is chosen to expand the interval between the two classes, whereby a contrastive loss function is set, as in Equation (3).

$$Loss_{Contras} = \begin{cases} \mathcal{S}_{pos} - \mathcal{S}_{neg} + \alpha, & \mathcal{S}_{pos} - \mathcal{S}_{neg} + \alpha > 0 \\ 0, & otherwise \end{cases} \quad (3)$$

For the hyper-parameter, we pre-set $\alpha = 0.01$.

### 3.3. Prediction with Class Activation Map (CAM)

Since the Siamese networks share parameters, one branch can be cropped in the application phase. As shown in Figure 3, after cropping the Siamese structure into a single branch structure, the original and segmentation images are concatenated as the input to the neural network to obtained the evaluation score. According to a previous study [55], semantic meaningful regions usually play an important role in deciding the segmentation quality. To better capture these regions and integrate the information into the quality calculation, a CAM is used to adjust the score. Considering performance, we use the smooth Grad-CAM++ method to obtain the CAM. On this basis, a FreqCAM [55] module is added.

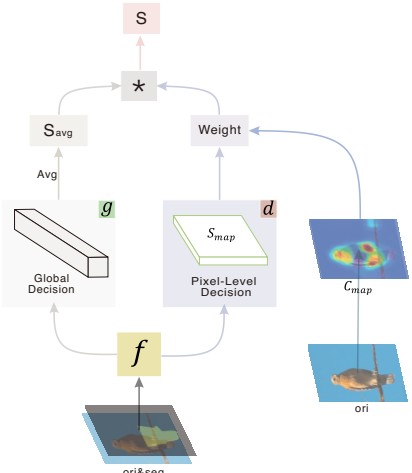

**Figure 3.** Prediction with a class activation map (CAM). The CAM is used as a weight to produce a weighted average, *Weight*, of the pixel-level matching results $S_{map}$. $*$ is the product operation. In this way, higher weights are assigned to attention regions.

FreqCAM [56] is a simple module for weakly supervised object localization, which gives higher weights to the attention region while eliminating most of the weight of the

background regions, especially the noise regions, in line with the intention of using a CAM in this study. Therefore, we apply FreqCAM to our no-reference segmentation quality evaluation scenario.

We define the original image as *ori*, the segmentation image as *seg* and the CAM as $C_{map}$. For pairs $(ori, seg)$, the feature $F$ is obtained by feeding it into the function $f$, the similarity matching map $S_{map}$ with $320 \times 320$ dimension is obtained by function $d$, and the $S_{avg}$ is obtained by the function $g$ and the averaging operation.

In the CAM, the attention region is assigned a higher weight. Therefore, using $C_{map}$ as the weight vector map and $S_{map}$ as the base vector map, a weighted average value *Weight* is computed to reflect the accuracy of the attention region, which is defined as Equation (4).

$$Weight = \frac{S_{map} * C_{map}}{\sum C_{map}} \tag{4}$$

Using *Weight* as a coefficient reflecting the accuracy of the attention component, a threshold penalty method is used for calculating the final score $S$. That is, a threshold is set, and when the *Weight* is higher than the threshold, the score is unchanged, and when it is lower than the threshold, the score is reduced. The coefficient is used directly as the disciplinary ratio. In this paper, we set $threshold = 0.5$. The final score $S$ is calculated by Equation (5).

$$Loss_{Contras} = \begin{cases} S_{avg}, & Weight > threshold \\ Weight * S_{avg}, & otherwise \end{cases} \tag{5}$$

Moreover, we consider adjusting the scores based on a threshold penalty method. This method keeps the scores with higher weights unchanged and only penalizes scores with lower weights.

As shown in Figure 4, if the segmentation score is adjusted corresponding to (1), the score will be too low. It would be more appropriate to only adjust the score for scenarios such as (2), keeping (1) unchanged. After comprehensive consideration, we set $threshold = 0.5$.

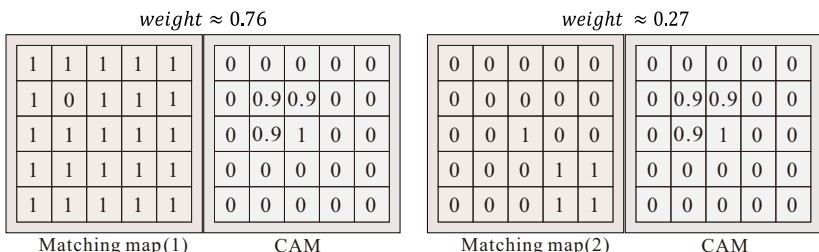

**Figure 4.** Scenarios of unsuitable and suitable adjustment scores. The matching map refers to the matching result of the original image and the segmentation image output in the pixel-level similarity learning stage. A CAM is generated to reflect the attention region. Scene (1) is not suitable for adjustment, because the matching map reflects the segmentation image and the original image has a high match, and adjusting it will result in too low a score. Scene (2) is suitable for tweaking.

## 4. Experiments

### 4.1. Experimental Configuration

Our method was trained and validated on an NVIDIA GeForce 2080Ti GPU with 11 GB memory, python3.7, and pytorch1.1. We use FCN with ResNet-50 as the backbone network. ResNet-50 was loaded as the pre-trained network parameter in the ImageNet dataset; the parameters of the first layer were not loaded because our structure changed from three channels to six channels. We set the Stochastic Gradient Descent (SGD) optimizer with a $4 \times 10^{-4}$ learning rate, 0.9 momentum, and 0.001 weight decay. The similarity learning and contrastive learning batch size was set to 16. The number of similarity learning epochs was 120 and for contrastive learning it was 5.

### 4.2. Dataset

In this paper, the original images and ground truth segmentation labels were selected from the Pascal Voc2012 dataset, where 20 foreground classes and 1 background class are included. Four SOA methods were chosen to produce the segmentation image sets, including FCN [25], U-Net [57], Mask-RCNN [58], and DeepLabV3 [59]. The segmentation results with 15 and 25 epochs for each method are selected, for a total of eight segmentations. Then, one each of good segmentation and poor segmentation samples were selected from the eight segmentation samples to make two segmentation image sets. Finally, a total of 8887 samples were used, using 7937 samples for training and 950 samples for validation, each sample containing four kinds of data: the original image set, the good segmentation image set, the poor segmentation image set, and the ground truth image set. All images were unified into dimensions $(H, W, C)$ of $(320, 320, 3)$.

### 4.3. Experimental Criteria

We verify multiple evaluation methods for both SISD and CISD meta-measures. SISD measures two segmentation datasets separately, a good segmentation and a poor segmentation dataset output by state-of-the-art (SoA) methods. The poor segmentation has some regions that are over-segmented, under-segmented, or misclassified, but still has a strong correlation with the original image and should be better than the segmentations generated by a different original image. CISD compares good segmentations and poor segmentations for the same original image.

#### 4.3.1. Meta-Measure for SISD

As shown in the left of Figure 5, SISD (Swapped-Image SoA Discrimination) compares the results created by a SoA segmentation method with the results created by the SoA segmentation method on other original images. For each SoA segmentation technique, SISD computes the number of images in the dataset in which an evaluation measure correctly judges that the corresponding SoA image result is better than the different image result. The definition of meta-measure **SISD (Swapped-Image SoA Discrimination)** is the percentage of results in the database that are correctly discriminated [18].

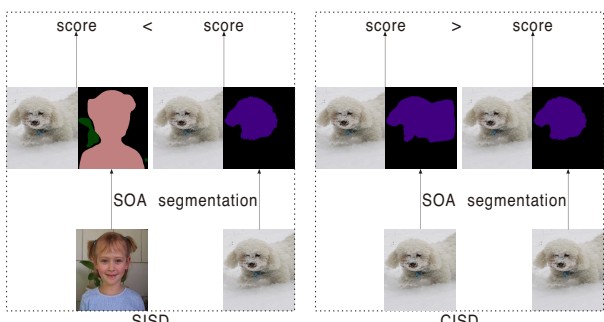

**Figure 5.** Illustration of SISD and CISD meta-measures. SISD compares the results created by SoA methods with the results created by the same SoA method but on other original images. CISD compares the good results and the bad results created by SoA methods for the same original image.

Specifically, SISD is defined in Equation (6), where $x_i$ refers to the $i$th original image, $y_i$ refers to the segmentation image for $x_i$, and function S () outputs the matching score between the original image and the segmentation image. $c_i$ is a judgement, and it is considered true when $S(x_i, y_i) > S(x_i, y_j)$. $S(x_i, y_j)$ is the matching score for $x_i$ and other segmentation images.

$$SISD = \frac{\sum_{i=0}^{n} c_i}{n}, \begin{cases} c_i = 1, & S(x_i, y_i) > S(x_i, y_j), i \neq j \\ c_i = 0, & otherwise \end{cases} \tag{6}$$

4.3.2. Meta-Measure for CISD

SISD meta-measure was performed only for the differentiation of swapped images; however, this is not enough. Most of the segmentation methods are validated on the same image; thus, a new meta-evaluation method is proposed in this paper, which is named **CISD (Corresponding Image SoA Discrimination)**, and is the percentage of correctly judged images from the good segmentation result of SoA that are more correlated with the corresponding images than the bad results (corresponding image refers to the original image corresponding to the good and poor segmentation image, that is, the segmentation image pair to be compared.). CISD is shown in the right of Figure 5.

Specifically, CISD is defined in Equation (7), where $x_i$ refers to the $i$th original image and $y_{i,pos}$, $y_{i,neg}$ refers to the segmentation image for $x_i$, where $y_{i,pos}$ has a higher quality than $y_{i,neg}$. Function S () outputs the matching score between the original image and the segmentation image. $c_i$ is a judgement, and it is considered true when $S(x_i, y_{i,pos}) > S(x_i, y_{i,neg})$.

$$SISD = \frac{\sum_{i=0}^{n} c_i}{n}, \begin{cases} c_i = 1, & S(x_i, y_{i,pos}) > S(x_i, y_{i,neg}) \\ c_i = 0, & otherwise \end{cases} \tag{7}$$

*4.4. Comparison of Meta-Measure Results of Our Method and Other Methods*

As shown in Table 1, SISD (good) represents the SISD measure in good segmentation sets and SISD (bad) represents the SISD measure in poor segmentation sets. The global accuracy is defined as the average of the above three evaluation methods, and the global accuracy of our method exceeds that of the no-reference method. In CISD, the traditional no-reference methods are somewhat effective, but less so than our method. In SISD, our method demonstrates absolute superiority, even over full-reference methods. Traditional no-reference methods are not very effective.

**Table 1.** Meta-measure results of different methods. SISD (good) refers to a comparison of the good segmentation results from SoA, and SISD (bad) is the comparison of poor results. CISD is the comparison of good and bad results in the same original image. Global is their average.

| Methods/Accuracy (%) | Global | CISD | SISD (Good) | SISD (Bad) |
|---|---|---|---|---|
| *Reference Methods* | | | | |
| *MPA* [9] | 90.05 | 78.26 | 97.29 | 95.81 |
| *PRI* [13] | 90.05 | 78.26 | 97.29 | 95.81 |
| *Precision* [25] | **93.45** | **81.62** | 99.65 | 99.22 |
| *Recall* [25] | 86.09 | 63.55 | 97.82 | 96.91 |
| *F-measure* [12] | 91.21 | 74.89 | 99.52 | 99.21 |
| *MIoU* [10] | 91.00 | 74.58 | 99.43 | 99.00 |
| *Dice* [14] | 90.78 | 74.00 | 99.40 | 98.94 |
| *VI* [26] | 91.02 | 74.63 | 99.43 | 99.00 |
| *No-reference Methods* | | | | |
| *F* [17] | 57.69 | 68.90 | 53.89 | 50.27 |
| *F'* [27] | 58.50 | 71.32 | 53.89 | 50.29 |
| *Q* [16] | 53.02 | 71.32 | 46.37 | 41.37 |
| *E* [15] | 38.07 | 38.74 | 49.75 | 25.71 |
| *Ecw* [29] | 45.42 | 44.53 | 47.96 | 43.78 |
| *Zeb* [28] | 49.18 | 71.87 | 8.18 | 67.49 |
| *Our* | **92.04** | **77.02** | **99.84** | **99.27** |

### 4.5. Comparison of the Evaluation Score of Our Method and Other Methods

To further verify the validity of our method, in addition to the meta-measures, we also show all the evaluation scores in the validation set for the reference methods and our method. Other no-reference methods are not shown because of their slow computation efficiency and irregular values.

In particular, as shown in Figure 6, for each original image displayed on the vertical axis, yellow and green are the mean of the evaluation scores of the segmentation images generated by all the other original images. It can be seen that our no-reference method is comparable with reference methods. MPA and Recall have high scores in other image segmentation results. Precision, F-measure, MIoU, Dice, and VI have evaluation scores that are completely separate from other image segmentation results, but too many low scores give the original image own segmentation results.

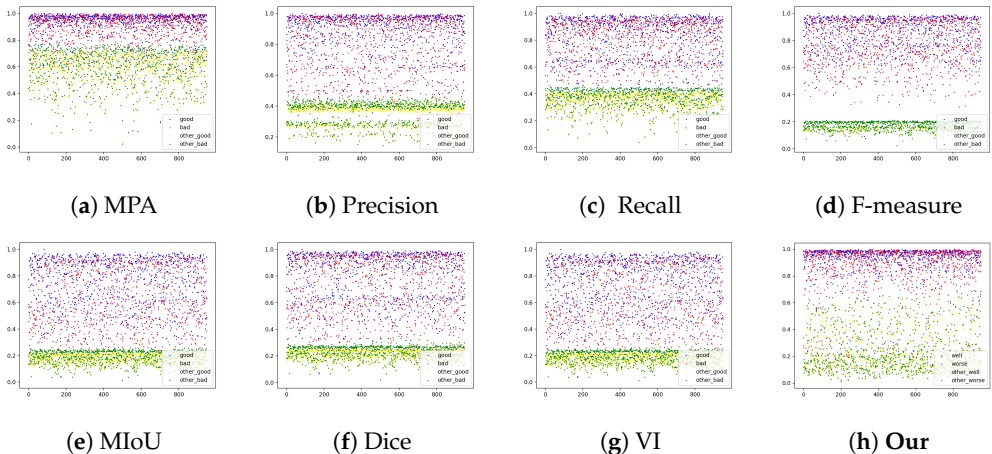

**Figure 6.** Comparison of the evaluation scores of reference methods and our method. The horizontal axis represents the sequence of images and the vertical axis represents the score. A blue point represents good segmentation evaluation scores of the original image itself generated by SoA methods, and a red point is a bad score. A yellow point represents the average value for the good segmentation evaluation score generated by all other original images, and a green point is a bad score.

### 4.6. Ablation Study

In this section, we analyze the effectiveness of our proposed method, performing ablation studies on each proposed component of the network architecture and empirically analyzing the corresponding reason. Ablation studies contain both the meta-measure accuracy and the details of different learning epochs. SL represents using only pixel-level similarity learning. CL represents contrastive learning, SCAM represents Smooth Grad-CAM++, FCAM represents FreqCAM, and 0.5, 0.75, and 1 represent the penalty thresholds.

#### 4.6.1. Compare Accuracy with the Addition of Different Module

As shown in Table 2, when using only similarity learning, the result is a pixel-level accuracy map. Instead of setting a threshold to predict the correctness or the incorrectness, a soft average [60] is used directly as the evaluation score. This method gives poor results as it only evaluates the pixel-level relationships. When feature contrastive learning is added, there is a large improvement in the accuracy, by 2.04 percentage points. Thus, feature contrastive learning is effective in this scenario. With the adoption of a CAM with different penalty thresholds, in meta-measures, the performance of our method was further improved. The best performance is achieved using FCAM accompanied by a threshold of 0.5. When the threshold is set to 1, it means that instead of using the threshold penalty method, the evaluation results of all segmentation images are adjusted by weight. In this case, it achieves a lower accuracy, which validates our assumption in Section 3.3.

**Table 2.** Ablation study for our method based on meta-measures. Contrastive learning leads to a big boost compared to just pixel-level similarity learning, and the addition of a CAM also results in a slight boost.

| Methods/Accuracy (%) | Global | CISD | SISD (Good) | SISD (Bad) |
|:---:|:---:|:---:|:---:|:---:|
| $Our_{SL}$ | 89.71 | 71.16 | 99.37 | 98.60 |
| $Our_{SL+CL}$ | 91.97 | 76.84 | 99.79 | 99.28 |
| $Our_{SL+CL+SCAM_1}$ | 91.70 | 76.11 | 99.78 | 99.22 |
| $Our_{SL+CL+FCAM_1}$ | 91.34 | 75.05 | 99.83 | 99.14 |
| $Our_{SL+CL+SCAM_{0.75}}$ | 91.88 | 76.53 | 99.82 | 99.28 |
| $Our_{SL+CL+FCAM_{0.75}}$ | 91.81 | 76.42 | 99.81 | 99.19 |
| $Our_{SL+CL+SCAM_{0.50}}$ | 92.02 | 76.95 | 99.81 | **99.30** |
| $Our_{SL+CL+FCAM_{0.50}}$ | **92.04** | **77.02** | **99.84** | 99.27 |

4.6.2. Details of Meta-Measure Results in Different Epochs

We show the training process and results in more detail in Figure 7, which contains the accuracy of the meta-measures over 120 epochs of pixel-level similarity learning and 5 epochs of contrastive learning. In order to show the differences between epochs more clearly, the *y*-axis is not consistent.

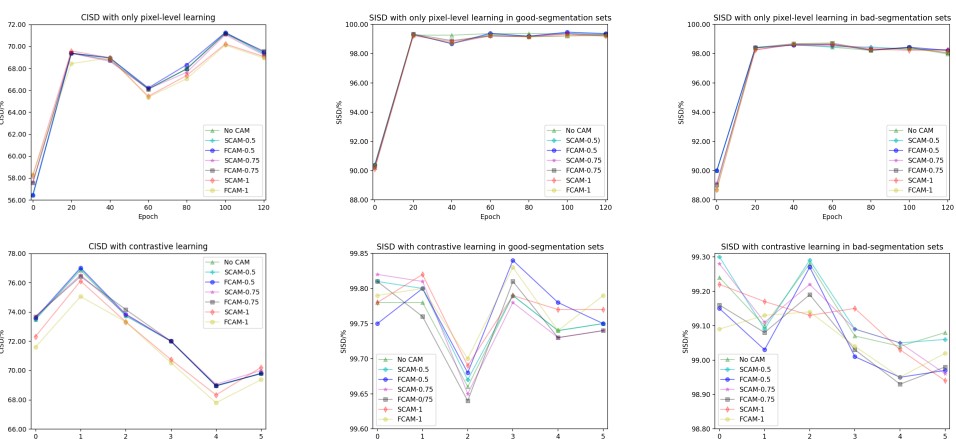

**Figure 7.** The SISD and CISD of different epochs. The upper part only uses the training process and results of pixel-level similarity learning, with 120 iterations in total. The lower part adds contrastive learning with five iterations.

*4.7. Evaluation Examples from Different Methods*

Figure 8 and Tables 3–5 show examples of different ways to evaluate scores. Figure 8 is the sample image selected. We have a selection of typical three full-reference methods and three no-reference methods for comparison. For each original image, we choose six segmentation images for evaluation, indicating good and poor segmentations as good and poor; good and bad segmentation images from different original images with close categories as other-good (1) and other-bad (1); and the good and poor segmentations from different original images that are close to the target location but in different categories as other-good (2) and other-bad (2).

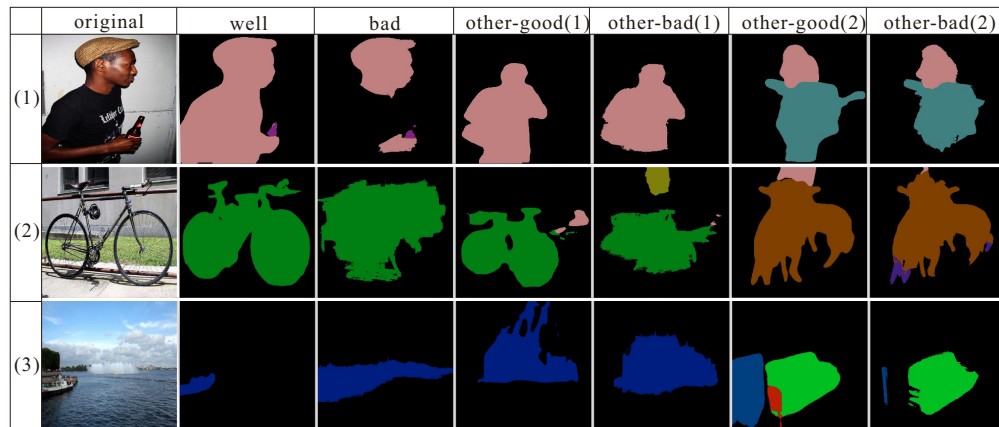

**Figure 8.** Example images selected for evaluation. We chose three original images as the base, and in part (1), part (2), and part (3), six segmentation images were selected for each original image to show the evaluation results. The six segmentation images include the good and bad segmentations generated by the original images, which are are denoted as good and bad. The good and poor segmentations generated from other original images include similar categories are denoted as other-good (1) and other-bad (1). The good and poor segmentations from different original images that are close to the target location but in different categories are denoted as other-good (2) and other-bad (2). Different colors represent different categories, such as pink is the people, green is the bicycle, and blue is the ship.

**Table 3.** Evaluation results of the examples in part (1) of Figure 8.

| Methods/Score (%) | Good | Bad | Other-Good (1) | Other-Bad (1) | Other-Good (2) | Other-Bad (2) |
|---|---|---|---|---|---|---|
| *MIoU* [10] | 0.76 | 0.43 | 0.44 | 0.40 | 0.18 | 0.18 |
| *Precision* [25] | 0.78 | 0.55 | 0.53 | 0.50 | 0.33 | 0.34 |
| *F-measure* [12] | 0.87 | 0.66 | 0.65 | 0.63 | 0.14 | 0.42 |
| *F* [17] | 0.17 | 0.30 | 0.09 | 0.19 | 0.18 | 0.29 |
| *E* [15] | 2.69 | 1.39 | 1.65 | 1.36 | 1.96 | 1.73 |
| *Ecw* [29] | 0.50 | 0.20 | 0.31 | 0.27 | 0.32 | 0.28 |
| *Ours* | 0.98 | 0.67 | 0.27 | 0.26 | 0.10 | 0.12 |

**Table 4.** Evaluation results of the examples in part (2) of Figure 8.

| Methods/Score (%) | Good | Bad | Other-Good (1) | Other-Bad (1) | Other-Good (2) | Other-Bad (2) |
|---|---|---|---|---|---|---|
| *MIoU* [10] | 0.21 | 0.22 | 0.19 | 0.15 | 0.12 | 0.11 |
| *Precision* [25] | 0.51 | 0.47 | 0.35 | 0.35 | 0.21 | 0.22 |
| *F-measure* [12] | 0.42 | 0.42 | 0.10 | 0.26 | 0.35 | 0.23 |
| *F* [17] | 0.28 | 1.54 | 0.66 | 1.83 | 0.26 | 0.44 |
| *E* [15] | 3.04 | 2.90 | 2.11 | 1.99 | 2.45 | 2.28 |
| *Ecw* [29] | 0.50 | 0.29 | 0.31 | 0.27 | 0.40 | 0.36 |
| *Ours* | 0.98 | 0.67 | 0.12 | 0.12 | 0.09 | 0.09 |

**Table 5.** Evaluation results of the examples in part (3) of Figure 8.

| Methods/Score (%) | Good | Bad | Other-Good (1) | Other-Bad (1) | Other-Good (2) | Other-Bad (2) |
|---|---|---|---|---|---|---|
| *MIoU* [10] | 0.78 | 0.48 | 0.38 | 0.38 | 0.13 | 0.20 |
| *Precision* [25] | 0.92 | 0.93 | 0.39 | 0.47 | 0.26 | 0.42 |
| *F-measure* [12] | 0.86 | 0.70 | 0.44 | 0.48 | 0.29 | 0.37 |
| *F* [17] | 0.05 | 0.74 | 0.38 | 0.85 | 0.61 | 0.29 |
| *E* [15] | 0.24 | 1.24 | 1.53 | 1.66 | 3.23 | 1.20 |
| *Ecw* [29] | 0.01 | 0.18 | 0.22 | 0.25 | 0.49 | 0.17 |
| *Ours* | 0.97 | 0.40 | 0.24 | 0.20 | 0.21 | 0.37 |

In Tables 3–5, it is important to note that the values for no-reference methods F, E, and Ecw represent errors, with higher scores indicating a poorer quality. For *F*, we multiplied the value by $10^3$, e.g., when the displayed value is 0.17, the actual evaluation value is $0.17 \times 10^{-3}$.

Table 3 shows the corresponding evaluation results of part (1). In part (1), the evaluation scores of the full-reference methods on the other-good (1) and other-bad (1) are too high, as they are evaluated based on the segmentation space only. Our method scores are more reasonable compared to other methods.

Table 4 shows the corresponding evaluation results of part (2). In part (2), the full-reference method has a good performance in other image segmentation results, but produces too low evaluation scores in the corresponding good and bad segmentations. The evaluation score of our method is more reasonable.

Table 5 shows the corresponding evaluation results of part (3). In part (3), due to the small foreground target, the full-reference method Precision and F-measure produced excessive scores in poor segmentation; our method overcomes this problem.

In summary, it can be seen that no-reference methods have difficulty comparing segmentation from the corresponding image with segmentation from other images, and the scores are also very unreasonable. In our method, three structures are included to evaluate the score, one using only pixel-level similarity learning, then adding contrastive learning, and finally adding a CAM. For the poor segmentations, i.e., good/bad segmentations from other images, our method does not produce reasonable scores in the first two methods, and along with the CAM, this score is more compatible with human vision than other methods.

## 5. Discussion

In this section, we discuss why the CAM does not exhibit a large accuracy improvement but is still adopted in this study. The reason for adding a CAM is that it makes the evaluation scores more close to human perception. We calculated the scores for all segmentation images for validation, as shown in Figure 9. A blue point represents a good segmentation image score for the corresponding original image, and a red point represents a bad one. A yellow point represents the evaluation of a good segmentation from a different original image, and a green point represents a bad one. By default, the CAM penalty threshold is set to 0.5. Obviously, if a CAM is not used, the segmentations generated by different original images are over-scored. In conjunction with a CAM, the scores of segmentations generated by different original images are suppressed and the scores are more reasonable.

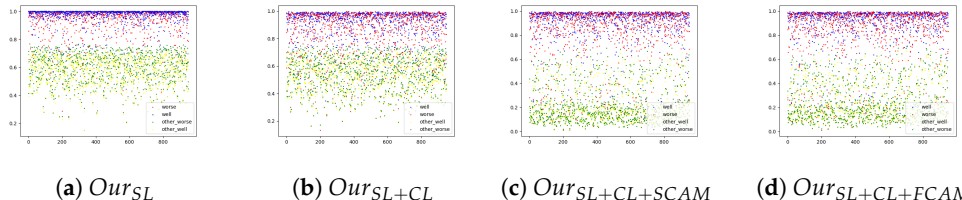

(a) $Our_{SL}$     (b) $Our_{SL+CL}$     (c) $Our_{SL+CL+SCAM}$     (d) $Our_{SL+CL+FCAM}$

**Figure 9.** Evaluation scores for the validation dataset with different modules. SL represents pixel-level similarity learning, CL represents contrastive learning, SCAM represents Smooth Grad-CAM++, and FCAM represents FreqCAM. The horizontal axis represents the sequence of images and the vertical axis represents the score. A blue point represents a good segmentation evaluation score of an original image itself generated by SoA methods, and a red point is a bad one. A yellow point represents the average value of the good segmentation evaluation scores generated by all other original images, and a green point is the bad one.

## 6. Conclusions

This work aims to improve the performance of no-reference segmentation evaluation methods. Among the currently existing no-reference methods, the performance of traditional methods is inadequate and few methods are based on deep learning. Moreover, the methods cannot be applied to semantic segmentation scenarios. We attempt to evaluate segmentation images using a learning method from the perspective of understanding the relationship between the original image and the segmentation image, and thus propose a feature contrastive learning method for no-reference segmentation quality evaluations. This method can be applied to semantic segmentation scenarios and is not worse than full-reference methods. In the experimental phase, we proposed the CISD meta-measure for validating the evaluation accuracy of different methods for segmentation of the corresponding image, and our method outperforms other no-reference methods. In the popular meta-measure SISD, our method outperforms the full-reference methods, which indicates its superiority in practical applications.

**Author Contributions:** Methodology, X.L.; writing—original draft, X.L.; writing—review and editing, B.P. and Z.X. All authors have read and agreed to the published version of the manuscript.

**Funding:** This work was supported by the Natural Science Foundation of Sichuan (no. 2022NS-FSC0502), Sichuan Science and Technology Program (nos. 2022ZYD0117 and 2023YFG0354) and the Key Research and Development Program of Sichuan Province (no. 2023YFG0125).

**Institutional Review Board Statement:** Not applicable.

**Informed Consent Statement:** Not applicable.

**Data Availability Statement:** The code and dataset link will released on: https://github.com/a6177738/Feature-Contrastive-Learning-for-NoReference-Segmentation-Quality-Evaluation.

**Conflicts of Interest:** The authors declare no conflict of interest.

## Abbreviations

The following abbreviations are used in this manuscript:

| | |
|---|---|
| CAM | Class Activation Map |
| SoA | State-of-the-Art |
| MIoU | Mean Intersection over Union |
| MPA | Mean Pixel Accuracy |
| PRI | Probabilistic Rand Index |
| Dice | Dice Coefficient |
| VI | Variation of Information |
| SISD | Swapped Image SoA Discrimination |

| CISD | Corresponding Image SoA Discrimination |
| CNN | Convolutional Neural Networks |
| KNN | K-Nearest Neighbor |
| SVM | Support Vector Machine |
| CL | Contrastive Learning |
| SL | Pixel-level Similarity Learning |
| SCAM | Smooth Grad-CAM++ |
| FCAM | FreqCAM |
| FCN | Fully Convolutional Networks |
| SGD | Stochastic Gradient Descent |

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
