# Peer review of "Feature Contrastive Learning for No-Reference Segmentation Quality Evaluation"

_electronics, doi:10.3390/electronics12102339_

Round 1

Reviewer 1 Report

The authors proposed a feature contrastive learning method for the segmentation of images without reference masks. Following are the comments that authors must address in the revised version of the paper.

1.     Remove section 0.

2.     Line 98: The third contribution of the paper is extensive testing to validate the proposed evaluation framework. So it is part of contributions 1 and 2.

3.     The name of section 3 should be Proposed Evaluation Framework.

4.     Line 222: Why H and W are set to 320? The C variable is not defined.

5.     Line 238: What is the reason for using the number 256 in H x W x256.

6.     All the abbreviations should be defined when used for the first time.

7.     Table 1: Define global accuracy.

8.     Meta-Measures for SISD and CISD are not adequately explained.

9.     Line 373: First, figure 6 should be explained in detail. The explanation of Figure 6 in lines 37-378 is not clear. The tables should be explained after Figure 6.

10.  Figure 6 shows well segmentation and bad segmentation. But it contains only one foreground object and the rest is background. Moreover, the last three columns other-well, other-bad, and other-bad represent what type of segmentation?

11.  Tables 2, 3, 4: Where is Figure 11?

12.   Line 386: The sentence “In part (1), the evaluation scores of the full-reference methods on the first other well-segmentation and first other bad-segmentation are too high, as they are evaluated based on the segmentation space only” is very confusing. Too many others are used.

13.  For multiple images, the evaluation results in Tables 2, 3, and 4 should be presented as mean and standard deviation. Using only three images for evaluation is not reasonable.

14.  Line 379: “In Table.2, Table.3 and Table.4, SL represents using only pixel-level similarity learning. CL represents contrastive learning, SCAM represents Smooth Grad-CAM++, FCAM represents FreqCAM”. In these tables, there are no CL, SL, or SCAM.

15.  In the dataset the authors divided the dataset into training and validation. Whereas in section 5.2, another testing dataset is not defined earlier.            

Extensive proofreading is required.

Reviewer 2 Report

This paper proposes a no-reference segmentation quality evaluation method based on contrastive learning and class activation map. Experiments show the superiority of the proposed method when compared to existing ones. Some other comments are:

1. Many details are missing for the proposed method, e.g., what is the global decision?

2. The results in Table 5 do not have explicit differences. Thus, the significant tests would be helpful to further validate the performance.

3. To provide reader with better understanding, except for the segmentation quality evaluation, traditional image quality evaluation and other quality evaluation like salient object detection can be added for references (to point out the differences between these tasks), including GraphIQA, metaIQA, LIQA: Lifelong blind image quality assessment, Structure-measure: a new way to evaluate foreground maps, etc.

4. It is suggested to improve the presentation. For example, section 0 can be removed. Contribution 3 is about experiments that can be merged into other contributions.

n/a

Reviewer 3 Report

The authors have submitted an interesting manuscript about the no-reference quality assessment of image segmentation. The topic of the paper is important nowadays and many possible applications can be enumerated. I think the abstract is good and provides enough information about the background of the proposed method and refers to the achieved results. In Line 3-4, the word "demanding" is probably not appropriate. Please consider "demand" instead. Section 0 "how to use this template?" should be deleted from the manuscript. I think that the Introduction section is generally and provides enough information about the background of the research topic. In Line 60-68, the authors could mention that contrastive learning is a popular topic in computer vision research with many applications, such as person reidentification (Person Re-identification based on Deep Multi-instance Learning, 2017), image matching (Siamese Network Features for Image Matching, 2016), or visual tracking (Structured Siamese Network for Real-Time Visual Tracking, 2018). In general, the authors use many abbreviations in the manuscript. This is why, a list of abbreviations at the end of the paper would be useful for readers. The related work section is good, although there is a chaotic sentence in Line 129. The reader first does not know what E, Q, and F refer in the text. Please consider to rewrite this part of the paper. Unfortunately, the proposed approach part of the paper is not so well written as the previous parts of the paper. Figure 1's caption is very short and not informative at all. In general, figure captions should be self-contained. Further, there are many typos in this part of the paper, such as "difine". It is very odd that the authors first give the exact value of H and W but in the later parts, they do not use these exact values but symbols are used. I think the exact values with explanations can be given at the end of the description of the proposed method or in a parameter study in the experimental results section. The caption of Figure 2 is also very short. Figure captions should be informative and self-contained, as already mentioned. Moreover, the structure of the CNN - denoted by f is Figure 2 - remained unclear to me. A separate figure about it would be probably useful to readers. Caption of Figure 3 is also very short and not informative at all. Figure 4 says almost nothing. Please draw a more informative figure. In the experimental results section, the definition of the evaluation metrics should be given. In Table 3, at least the best and second best values should be denoted (with colors or boldface). Since deep learning involves a lot of experiments, the publication of training curves would be nice. 

There are some typos in the text. Please check the text with a spell checking program and correct the typos.

Round 2

Reviewer 1 Report

The Authors have revised the paper and incorporated all my comments.

Reviewer 2 Report

The authors have addressed my comments.

n/a

Reviewer 3 Report

The authors have rewritten those parts of the manuscript where I had concerns or questions. The presentation of the experimental was also improved. I recommend this manuscript for publication.